# Learning-Augmented Online Bipartite Fractional Matching

**Davin Choo**[*][§]
Harvard John A. Paulson School Of Engineering And Applied Sciences
Harvard University
Boston, Massachusetts, USA
davinchoo@seas.harvard.edu

**Billy Jin**[†][§]
Daniels School of Business
Purdue University
West Lafayette, Indiana, USA
jin608@purdue.edu

**Yongho Shin**[‡][§]
Institute of Computer Science
University of Wrocław
Wrocław, Poland
yongho@cs.uni.wroc.pl

## Abstract

Online bipartite matching is a fundamental problem in online optimization, extensively studied both in its integral and fractional forms due to its theoretical significance and practical applications, such as online advertising and resource allocation. Motivated by recent progress in learning-augmented algorithms, we study online bipartite fractional matching when the algorithm is given advice in the form of a suggested matching in each iteration. We develop algorithms for both the vertex-weighted and unweighted variants that provably dominate the naïve "coin flip" strategy of randomly choosing between the advice-following and advice-free algorithms. Moreover, our algorithm for the vertex-weighted setting extends to the AdWords problem under the small bids assumption, yielding a significant improvement over the seminal work of Mahdian, Nazerzadeh, and Saberi (EC 2007, TALG 2012). Complementing our positive results, we establish a hardness bound on the robustness-consistency tradeoff that is attainable by any algorithm. We empirically validate our algorithms through experiments on synthetic and real-world data.

## 1 Introduction

Online bipartite matching is a fundamental problem in online optimization with significant applications in areas such as online advertising [MSVV07, FKM+09], resource allocation [DJSW19], and ride-sharing platforms [DSSX21, FNS24]. In its classical formulation [KVV90, AGKM11], the input is a bipartite graph where one side of (possibly weighted) *offline* vertices is known in advance, while the other side of *online* vertices arrives sequentially one at a time. When an online vertex $v$ arrives, its incident edges are revealed, and the algorithm irrevocably decides whether to match $v$ and, if so, to which currently unmatched neighbor. The objective is to maximize the total weight of the matched offline vertices. Algorithms for online bipartite matching are often evaluated by their *competitive ratio*: An algorithm is $\rho$-*competitive* if it always outputs a matching whose (expected) total weight is

---

[*]Part of work done while the author was affiliated with the National University of Singapore, Singapore.

[†]Work done while the author was at the University of Chicago Booth School of Business, USA.

[‡]Part of work done while the author was affiliated with Yonsei University, South Korea.

[§]Equal contribution.

at least $\rho$ times the weight of the best matching in hindsight. In a seminal paper, [KVV90] proposed the RANKING algorithm and showed it is $(1 - 1/e)$-competitive for the unweighted setting. This competitive ratio is best-possible, and was later extended to the vertex-weighted case by [AGKM11].

Online bipartite matching has also been studied in the fractional setting, where edges can be fractionally chosen, provided that the total fractional value on the edges incident to any vertex does not exceed one [WW15, HPT+19, HTWZ20, HHIS24]. Fractional matching is important both theoretically and practically. It naturally models settings where online arrivals are divisible or offline vertices have large capacities [KP00, MSVV07, BJN07, FKM+09, MNS12, DHK+16, FN24], and it forms the basis for designing integral algorithms using rounding techniques [FSZ16, BNW23, NSW25]. For fractional vertex-weighted online bipartite matching, the BALANCE algorithm of [BJN07] gets a competitive ratio of $(1 - 1/e)$, which is best-possible and matches the ratio in the integral case.

The main challenge in online bipartite matching is that irrevocable decisions must be made without knowledge of future arrivals. Uncertainty in the arrival sequence is typically modeled either *adversarially* or *stochastically*. The adversarial model assumes no structure and measures worst-case performance, but can be overly pessimistic. On the other hand, the stochastic model assumes arrivals are drawn from a known distribution [FMMM09], but such distributions are often estimated and may be inaccurate. These models thus represent two extremes, each with practical limitations. A middle ground is offered by *algorithms with predictions*, or *learning-augmented algorithms* [MNS12, LV21], which incorporates advice – derived from data, forecasts, or experts – of unknown quality. The performance is typically measured in terms of its *robustness* (guaranteed performance regardless of advice quality) and *consistency* (performance when advice is accurate) [LV21, KPS18].[5] In online bipartite matching, an algorithm is $r$-robust if its competitive ratio is at least $r$, and $c$-consistent if it achieves at least a $c$-fraction of the total weight from following the advice (see Definition 4). A natural baseline is the COINFLIP algorithm, which randomly chooses between robustness- and consistency-optimal strategies. For matching, its tradeoff curve is the line segment between $(1 - 1/e, 1 - 1/e)$ and $(0, 1)$ in the vertex-weighted case, or $(1/2, 1)$ in the unweighted case [JM22].

This paper investigates the robustness-consistency tradeoff of online bipartite matching under the learning-augmented framework, building on prior work including [MNS07, MNS12, ACI22, JM22, SE23, CGLB24]. Particularly relevant are the works of Mahdian et al. [MNS07, MNS12] and Spaeh and Ene [SE23]. Mahdian et al. studied the AdWords problem (introduced in [MSVV07]), with advice in the form of a recommendation assigning each online impression to a specific offline advertiser. They proposed a learning-augmented algorithm under the *small bids* assumption that outperforms the naïve COINFLIP strategy, but only over part of the robustness range. Meanwhile, [SE23] generalized this result to Display Ads and the generalized assignment problem [FKM+09]. However, as shown in Fig. 1, neither of these algorithms dominate COINFLIP across the full robustness spectrum. This raises a natural question: *Does there exist a learning-augmented algorithm for online bipartite matching that dominates* COINFLIP *across the entire range of robustness?*

## 1.1   Our contributions

We answer the above question affirmatively by presenting learning-augmented algorithms for both vertex-weighted and unweighted online bipartite fractional matching whose robustness-consistency tradeoffs Pareto-dominate that of COINFLIP across the *entire* range of robustness (see Fig. 1).

Motivated by [MNS07, MNS12, SE23], we take the advice to be a feasible fractional matching that is revealed in an online fashion: upon arrival of each online vertex $v$, the algorithm is given as advice fractional matching values for each neighboring edge of $v$. Moreover, as in [MNS07, MNS12, SE23], our algorithms are parameterized by a tradeoff parameter $\lambda \in [0, 1]$ that represents how closely we follow the advice. At the extremes, our algorithms blindly follow the advice when $\lambda = 1$ and revert to BALANCE when $\lambda = 0$.

For the vertex-weighted setting, we present an algorithm LEARNINGAUGMENTEDBALANCE (LAB) with the following guarantees:

**Theorem 1.** *For any tradeoff parameter* $\lambda \in [0, 1]$*,* LEARNINGAUGMENTEDBALANCE *is an* $r(\lambda)$*-robust and* $c(\lambda)$*-consistent algorithm for vertex-weighted online bipartite fractional matching, where*

$$r(\lambda) := 1 - e^{\lambda-1} - \left(e^{\lambda-1} - \lambda\right) \ln(1 - \lambda e^{1-\lambda}) - \lambda(1 - \lambda) \quad and \quad c(\lambda) := 1 + \lambda - e^{\lambda-1}.$$

---

[5]A third property, *smoothness*, requires graceful degradation with advice quality [EADL24]. See Section 2.

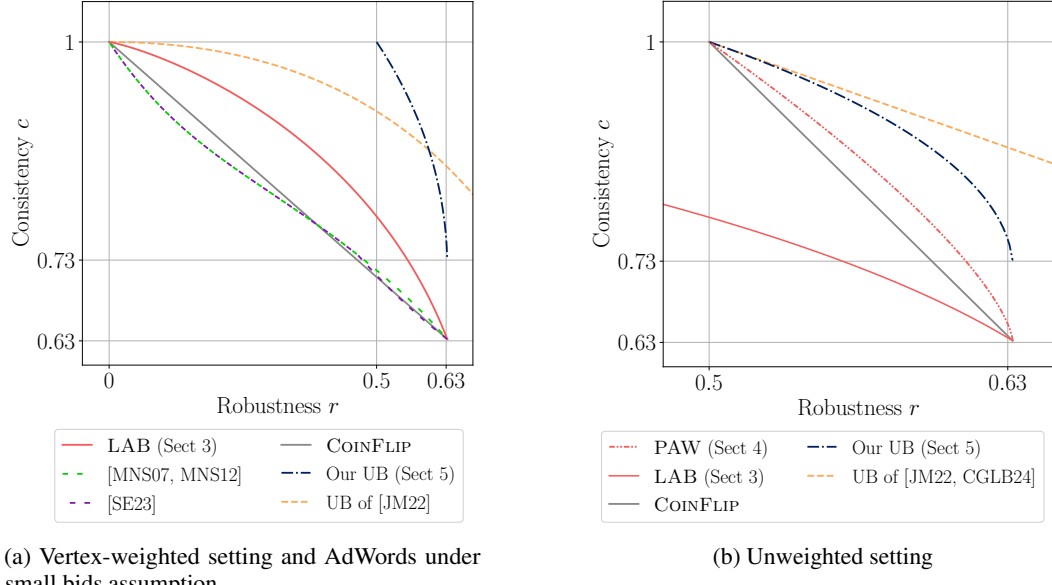

(a) Vertex-weighted setting and AdWords under small bids assumption

(b) Unweighted setting

Figure 1: Robustness-consistency tradeoffs of previous works and our results.

This algorithm is based on BALANCE where the penalty function is modified to be *advice-dependent*. To analyze this algorithm, we adopt the standard primal-dual analysis of online bipartite matching and first prove its performance when the advice is integral. We then prove that the robustness and consistency are minimized when the advice is integral, yielding the same guarantees for the general fractional advice case.

We further show that LAB extends to the AdWords problem under the small bids assumption, yielding a significant improvement over [MNS07, MNS12].

**Theorem 2.** *Consider the small bids assumption where the maximum bid-to-budget ratio is bounded by some sufficiently small $\varepsilon > 0$. For any tradeoff parameter $\lambda \in [0, 1]$, there exists an $r(\lambda) \cdot (1 - 3\sqrt{\varepsilon \ln(1/\varepsilon)})$-robust and $c(\lambda) \cdot (1 - 3\sqrt{\varepsilon \ln(1/\varepsilon)})$-consistent algorithm for AdWords with advice, where $r(\lambda)$ and $c(\lambda)$ are the same as in Theorem 1.*

To achieve this result, we first extend LAB to the *fractional* AdWords setting while preserving its robustness and consistency, and then employ a reduction from [FN24] to reduce the integral AdWords problem to the fractional problem with small loss under the small bids assumption.

Observe in Fig. 1 that the robustness-consistency tradeoff of LAB lies below the linear tradeoff of COINFLIP in the *unweighted* setting: the top-left endpoint of the tradeoff for LAB is $(r, c) = (0, 1)$, whereas in the unweighted setting COINFLIP can be implemented to be $1/2$-robust even when $c = 1$. This happens because any maximal matching in an unweighted graph is automatically $\frac{1}{2}$-robust. To beat COINFLIP in the unweighted setting, a tighter analysis of LAB would be required but this proved difficult using our current analysis framework for LAB, even when the advice is integral. Instead, we present another algorithm called PUSHANDWATERFILL (PAW) for the unweighted setting with integral advice that circumvents the aforementioned challenge in the analysis.

**Theorem 3.** *For any tradeoff parameter $\lambda \in [0, 1]$, PUSHANDWATERFILL is $r(\lambda)$-robust and $c(\lambda)$-consistent for unweighted online bipartite fractional matching with integral advice, where*

$$r(\lambda) := 1 - \left(1 - \lambda + \lambda^2/2\right) e^{\lambda-1} \ \ \text{and} \ \ c(\lambda) := 1 - (1 - \lambda) e^{\lambda-1}.$$

PAW is based on the unweighted version of BALANCE, also known as WATERFILLING, with one additional step at each iteration where it first increases the fractional value of the currently advised edge until the "level" of the advised offline vertex reaches the tradeoff parameter $\lambda$. We analyze PAW using primal-dual but with a different construction of dual variables from LAB.

We complement our algorithmic results by presenting an upper bound on the robustness-consistency tradeoff of any learning-augmented algorithm for the unweighted setting with integral advice in

Section 5, improving upon the previous upper bound results [JM22, CGLB24] (see Fig. 1). Note that this result implies the same impossibility for more general problems including the vertex-weighted setting and the AdWords problem. To obtain our hardness result, we construct two adaptive adversaries — one for robustness and the other for consistency. The construction of these adversaries is inspired by the standard upper-triangular worst-case instances [KVV90], while we modify this construction to make the two adversaries have the same behavior until the first half of the online vertices are revealed. Due to this modification, the two adversaries are indistinguishable until the halfway point of the execution while inheriting the difficulty from the standard worst-case instances. We then identify a set of conditions characterizing the behavior of Pareto-optimal algorithms on our hardness instance and solve a factor-revealing LP to upper bound the best possible consistency subject to the constraint on the robustness to be $r$, for each $r \in [1/2, 1 - 1/e]$.

Lastly, we implemented and evaluated our proposed algorithms LAB and PAW in Section 6 against advice-free baselines on synthetic and real-world graph instances, for varying advice quality parameterized by a noise parameter $\gamma$, where larger $\gamma$ indicates poorer advice quality. As predicted by our analysis, the attained competitive ratios of both LAB and PAW begin at 1 under perfect advice and smoothly degrades as the $\gamma$ increases. Unsurprisingly, for sufficiently large $\gamma$, the worst case optimal advice-free algorithm BALANCE outperforms both LAB and PAW.

Full proofs, further related work, and code are provided in the supplementary material. Environment numberings are made to be consistent with the full version, but may not appear sequentially here.

## 2 Preliminaries

**Online bipartite matching.** In the *vertex-weighted online bipartite fractional matching* problem, we have a bipartite graph $G = (U \cup V, E)$ and a weight $w_u \geq 0$ for each $u \in U$. If $w_u = 1$ for every $u \in U$, then the problem is called *unweighted*. The vertices in $U$ are the *offline* vertices, and their weights are known to the algorithm from the very beginning. On the other hand, the vertices in $V$ are the *online* vertices, and arrive one by one. Whenever $v \in V$ arrives, its neighborhood $N(v) := \{u \in U \mid (u, v) \in E\}$ is revealed. Since the online vertices arrive sequentially, we use the notation $t \prec v$ to mean that $t$ arrives earlier than $v$. Similarly, for each offline vertex $u \in U$, we also use $N(u) := \{v \in V \mid (u, v) \in E\}$ to denote the neighborhood of $u$.

We use the analogy of *waterfilling* to describe the behavior of the algorithm. When $v \in V$ arrives and its neighborhood $N(v)$ is revealed, the algorithm decides at that moment the amount $x_{u,v}$ of water to send from $v$ to each $u \in N(v)$ subject to the constraints that:

- the total amount of water supplied from $v$ does not exceed 1, i.e., $\sum_{u \in N(v)} x_{u,v} \leq 1$;
- each offline vertex $u \in U$ can hold at most 1 unit of water, i.e., $\sum_{t \in N(u):t \preceq v} x_{u,t} \leq 1$.

This decision is irrevocable, meaning that, $\{x_{u,v}\}_{u \in N(v)}$ cannot be modified in the subsequent iterations. Let $x \in \mathbb{R}^E$ be the final solution of the algorithm. Note that $x$ is a fractional matching in the hindsight graph $G$. The weight of this solution is defined to be $\sum_{(u,v) \in E} w_u x_{u,v}$. The objective of this problem is to maximize the weight of the solution.

**Advice.** Each online vertex $v \in V$ arrives with a *suggested allocation* $\{a_{u,v}\}_{u \in N(v)}$, where we assume $a = \{a_{u,v} : (u, v) \in E\} \in \mathbb{R}^E$ is a feasible fractional matching in the hindsight graph $G$.

**Performance measures.** Denote the value of the final output of an algorithm by ALG, the value of an optimal solution in the hindsight instance by OPT, and the value obtained by the advice by ADVICE. We can then formally define the *robustness* and *consistency* of a learning-augmented algorithm.

**Definition 4** (Robustness and Consistency). For some $r \in [0, 1]$, we say an algorithm is *r-robust* if $\mathbb{E}[\text{ALG}] \geq r \cdot \text{OPT}$ for any instance of the problem. On the other hand, for some $c \in [0, 1]$, we say an algorithm is *c-consistent* if $\mathbb{E}[\text{ALG}] \geq c \cdot \text{ADVICE}$ for any instance of the problem.

Notice that, when we define the *error* of the advice to be $\eta := \text{ADVICE}/\text{OPT} \in [0, 1]$, the consistency implies the *smoothness* of the algorithm since we have $\mathbb{E}[\text{ALG}] \geq c\eta \cdot \text{OPT}$.

**Primal-dual analysis.** To prove the robustness and consistency of our algorithms, we adopt the standard primal-dual analysis for online bipartite matching [DJK13]. Observe that, for vertex-weighted bipartite matching, the primal and dual LPs are formulated as follows:

$$\max \ \sum_{(u,v)\in E} w_u x_{u,v} \qquad\qquad \min \ \sum_{u\in U}\alpha_u + \sum_{v\in V}\beta_v$$

$$\text{s.t. } \sum_{v\in N(u)} x_{u,v} \le 1, \quad \forall u\in U, \qquad \text{s.t. } \alpha_u + \beta_v \ge w_u, \quad \forall (u,v)\in E,$$

$$\sum_{u\in N(v)} x_{u,v} \le 1, \quad \forall v\in V, \qquad \alpha_u \ge 0, \quad \forall u\in U,$$

$$x_{u,v} \ge 0, \quad \forall (u,v)\in E; \qquad \beta_v \ge 0, \quad \forall v\in V.$$

The following lemma is the cornerstone of the primal-dual analysis.

**Lemma 5** (see, e.g., [DJK13, FHTZ22])**.** *Let $x \in \mathbb{R}_+^E$ be a feasible fractional matching output by an algorithm. For some $\rho \in [0,1]$, if there exists $(\alpha,\beta) \in \mathbb{R}_+^U \times \mathbb{R}_+^V$ satisfying*

- *(reverse weak duality) $\sum_{(u,v)\in E} w_u x_{u,v} \ge \sum_{u\in U}\alpha_u + \sum_{v\in V}\beta_v$ and*

- *(approximate dual feasibility) $\alpha_u + \beta_v \ge \rho \cdot w_u$ for every $(u,v) \in E$,*

*we have $\mathsf{ALG} \ge \rho \cdot \mathsf{OPT}$.*

## 3 Vertex-weighted matching with advice

We now present our algorithm LEARNINGAUGMENTEDBALANCE (LAB) for vertex-weighted online bipartite matching with advice and provide a proof sketch showing that it achieves the robustness-consistency tradeoff stated in Theorem 1. Detailed pseudocode is given in Appendix A and a full analysis is provided in the supplementary material.

**Algorithm description.** Given a tradeoff parameter $\lambda \in [0,1]$, we define $f_0 : [0,1] \to [0,1]$ and $f_1 : [0,1] \to [0,1]$ as follows, where $W$ is the Lambert $W$ function:

$$f_0(z) := \min\{e^{z+\lambda-1}, 1\}, \quad \text{and} \quad f_1(z) := \begin{cases} \frac{e^{\lambda-1}-\lambda}{1-z}, & \text{if } z \in [0, \lambda e^{1-\lambda}), \\ \frac{-\lambda}{W(-\lambda e^{1-\lambda-z})}, & \text{if } z \in [\lambda e^{1-\lambda}, 1), \\ 1, & \text{if } z = 1, \end{cases} \tag{1}$$

Based on these functions, we define $f : [0,1]^2 \to [0,1]$ such that

$$f(A, X) := \begin{cases} f_1(X), & \text{if } A > X, \\ \max\{f_0(X-A), f_1(X)\}, & \text{if } A \le X. \end{cases} \tag{2}$$

For clarity, let us describe LAB as a continuous process. Upon the arrival of each online vertex $v \in V$ along with the advice $\{a_{u,v}\}_{u\in N(v)}$, define $A_u := \sum_{t\in N(u):t\preceq v} a_{u,t}$ as the total advice-allocated amount to each offline vertex $u \in N(v)$, up to and including $v$. LAB then continuously pushes an infinitesimal unit of flow from $v$ to the neighbor $u \in N(v)$ maximizing $w_u(1 - f(A_u, X_u))$, where $X_u$ is the total amount allocated to $u$ by the algorithm right before it starts pushing this infinitesimal unit of flow. This continues until $v$ is fully matched (i.e. one unit of flow is pushed) or all neighbors are saturated.

First, we give intuition for the algorithm. For an online vertex $v$ and an offline neighbor $u \in N(v)$, the amount allocated from $v$ to $u$ should depend on three factors. Firstly, a higher $w_u$ should lead to larger $x_{u,v}$. Secondly, the more $u$ is filled, the less desirable it is to allocate to it further, preserving capacity for future vertices. Thirdly, vertices favored by the advice should receive more allocation.

The classical BALANCE algorithm handles the first two factors by choosing the offline vertex with the highest potential value $w_u(1 - g(X_u))$ via a convex increasing penalty function $g(z) = e^{z-1}$. To incorporate the third factor, LAB introduces an advice-aware penalty function $f(A, X)$ that also depends on the total advice allocation $A$; see Fig. 2. This function is increasing in $X$ (penalizing already-filled vertices) and decreasing in $A$ (lower penalty for vertices recommended by the advice), thereby encouraging alignment with the advice.

The penalty function $f$ used by our algorithm is defined in Eq. (2) based on the functions $f_0$ and $f_1$ from Eq. (1). While $f_0$ and $f_1$ are derived from the primal-dual analysis, and their exact forms are not crucial for intuition, the structure of $f$ admits a natural interpretation. Intuitively, if an offline vertex

$u$ has received less allocation than the advice suggests (i.e., $A_u > X_u$), then the penalty function treats $u$ as if it were already saturated under the advice. Conversely, if $u$ has been filled beyond the advised amount (i.e., $A_u \leq X_u$), then the penalty effectively treats the excess allocation $X_u - A_u$ as if it were added despite the advice indicating $u$ should be unmatched.

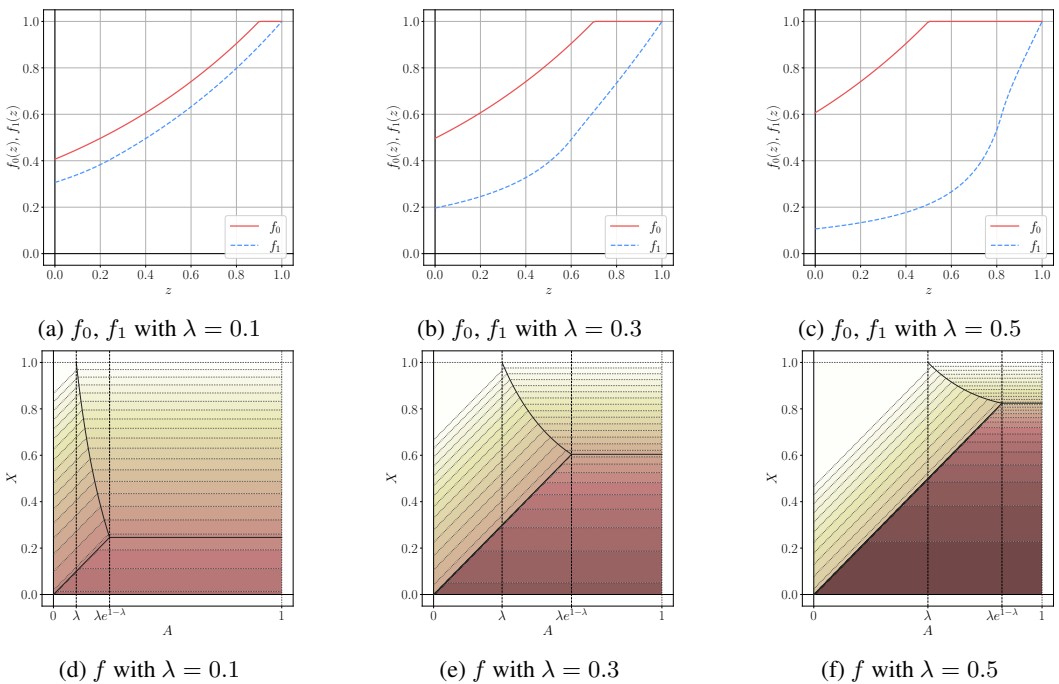

(a) $f_0$, $f_1$ with $\lambda = 0.1$     (b) $f_0$, $f_1$ with $\lambda = 0.3$     (c) $f_0$, $f_1$ with $\lambda = 0.5$

(d) $f$ with $\lambda = 0.1$     (e) $f$ with $\lambda = 0.3$     (f) $f$ with $\lambda = 0.5$

Figure 2: $f_0$, $f_1$, and $f$ with $\lambda \in \{0.1, 0.3, 0.5\}$. (a)-(c) depict the function values of $f_0$ and $f_1$ with respect to $z \in [0,1]$. (d)-(f) depict the contour plots with respect to $A \in [0,1]$ and $X \in [0,1]$: the brighter the color is, the closer to 1 the function value is.

**Sketch of analysis.** We use the primal-dual framework (Lemma 5). We initialize all dual variables to 0. When an online vertex $v \in V$ arrives, let $A_u^{(v)}$ be the amount allocated to $u$ by the advice up to and including $v$. Similarly, let $X_u^{(v)}$ be the amount allocated to $u$ by the algorithm up to and including $v$. Let $x_{u,v}$ be the amount allocated by the algorithm in this iteration.

Then, we update the dual variables as follows:

- $\alpha_u \leftarrow \alpha_u + x_{u,v} \cdot w_u f(A_u^{(v)}, X_u^{(v)})$ for every $u \in N(v)$, and
- $\beta_v \leftarrow \max_{u \in N(v)} \{ w_u (1 - f(A_u^{(v)}, X_u^{(v)})) \}$.

For the reverse weak duality, the following lemma follows directly from this construction:

**Lemma 9.** *The value of the algorithm is equal to the objective value of $(\alpha, \beta)$ in the dual LP.*

Note that the dual variable $\beta_v$ is defiend to be the value of the highest potential neighbor after $v$ has sent its allocation while the dual variables $\alpha_u$ are updated so as to satisfy Lemma 9.

We now argue the approximate dual feasibility. As a first step, we consider the special case where the advice is integral. In this case, we can simplify the robustness and consistency expressions by noting that $A_u^{(v)} \in \{0, 1\}$ for all $u \in U$ and $v \in V$.

**Lemma 12.** *When advice is integral, the algorithm is $r$-robust and $c$-consistent where*

$$ r = \min_{X \in [0,1]} \min \left\{ \int_0^X f_0(z)dz + (1 - f_0(X)), \int_0^X f_1(z)dz + (1 - f_1(X)) \right\} $$

$$ c = \min_{X \in [0,1]} \min_{Y \in [0,X]} \left\{ \int_0^Y f_0(z)dz + (X - Y) \cdot f_1(X) + (1 - f_1(X)) \right\} $$

*In addition, the expressions on the RHS above evaluate to exactly $r(\lambda)$ and $c(\lambda)$ in Theorem 1, respectively.*

To complete the analysis, one can show that robustness and consistency remain unchanged even for the more general fractional advice setting: robustness is determined entirely by $f_1$, while consistency follows from reducing any fractional instance to an equivalent integral case.

*Remark.* By tweaking the above analysis and using the reduction technique of [FN24], we can obtain an algorithm for AdWords under the small bids assumption with guarantees given in Theorem 2. For the details, see the supplementary material.

## 4   Unweighted matching with integral advice

In this section, we introduce and analyze a new algorithm tailored to the unweighted setting with integral advice, which we call PUSHANDWATERFILL (PAW). To motivate why we need a new algorithm, it is worth noting that our theoretical guarantees of LAB are dominated by COINFLIP in the unweighted setting; see Fig. 1. This suggests that our previous analysis is not tight for unweighted instances. However, since that analysis was independent of the vertex weights, we find it challenging to improve it for the unweighted setting, even when we are given integral advice. As such, we propose PAW for the setting of unweighted matching with integral advice. In the following, we assume the advice is integral and represent it as a function $A : V \to U \cup \{\bot\}$, where $A(v)$ is the advised match for $v \in V$, and $A(v) = \bot$ indicates that $v$ is advised to remain unmatched. Detailed pseudocode is given in Appendix A and a full analysis is provided in the supplementary material.

**Algorithm description.**   As before, we describe PAW as a continuous-time process. Define the level of an offline vertex $u \in U$ as the total amount of water it has received so far. Upon arrival of online $v \in V$, with neighborhood $N(v)$ and advice $A(v)$, the algorithm proceeds in two phases:

**Phase 1 (Push)**: Push flow into $A(v)$ until its level reaches $\lambda$.

**Phase 2 (Waterfill)**: Distribute any remaining flow from $v$ across $N(v)$ via the standard waterfilling.

**Sketch of analysis.** As in the analysis of LAB, we use a primal-dual framework to characterize the robustness and consistency of PAW. The dual variable construction differs from the vertex-weighted case and relies on a continuous and non-decreasing function $g : [0, 1] \to [0, 1]$ such that $g(1) = 1$. We call such a function a *splitting function*.

The dual variables $(\alpha, \beta)$ are initialized to zero, and are updated as follows. When an online vertex $v$ sends an infinitesimal amount $dz$ of flow to a neighbor $u \in N(v)$ whose current level is $d_u$, split this $dz$ into $g(d_u) \, dz$ and $(1 - g(d_u)) \, dz$. Then, we increase $\alpha_u$ by $g(d_u) \, dz$ and $\beta_v$ by $(1 - g(d_u)) \, dz$.

Since $g(d_u) \in [0, 1]$, both $\alpha$ and $\beta$ remain nonnegative. Moreover, by construction, the reverse weak duality in Lemma 5 holds with equality: every infinitesimal unit of flow is split *exactly* into two values contributing to $\alpha_u$ and $\beta_v$, respectively. Thus, to analyze the robustness and consistency of PAW, it suffices to identify splitting functions $g$ such that the resulting $(\alpha, \beta)$ is approximately dual feasible, maximizing the respective performance parameter $\rho$ in Lemma 5. To this end, we identify the following two splitting functions $g_r$ and $g_c$, tailored for robustness and consistency, respectively:

$$g_r(z) := \begin{cases} e^{\lambda-1}(z + 1 - \lambda), & \forall z \in [0, \lambda), \\ e^{z-1}, & \forall z \in [\lambda, 1] \end{cases} \quad \text{and} \quad g_c(z) := \begin{cases} e^{\lambda-1}, & \forall z \in [0, \lambda), \\ e^{z-1}, & \forall z \in [\lambda, 1]. \end{cases}$$

These functions lead to the robustness and consistency bounds in Theorem 3.

## 5   Upper bound on robustness-consistency tradeoff

In this section, we present an upper bound result for the unweighted setting with integral advice. We define two adversaries, $\mathcal{R}$ and $\mathcal{C}$, which target robustness and consistency, respectively, against any fractional matching algorithm $\mathcal{M}$. For a given positive integer $n$, both adversaries construct a bipartite instance with a set $U$ of $2n$ offline vertices and a set $V$ of $2n$ online vertices. See Fig. 6 for an illustration of the upper bound instance. Detailed pseudocodes of these adversaries and a full analysis are provided in the supplementary material.

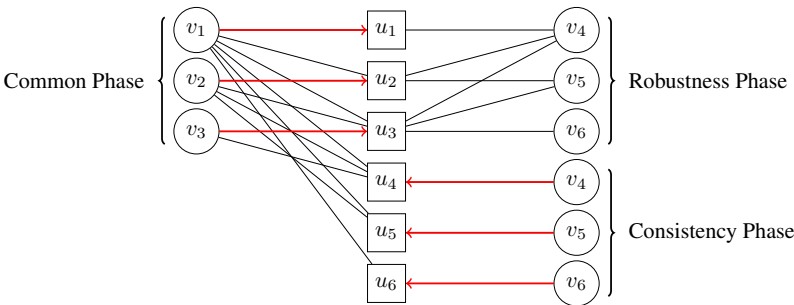

Figure 6: An illustration of the hardness construction. The instance begins with the common phase, which is the same in both adversaries. After the common phase, the instance can proceed in one of two ways, designed to be hard for robustness or consistency, respectively.

The two adversaries behave identically during the first $n$ iterations, as follows: In the first iteration ($t = 1$), they present the first online vertex $v_1$ to $\mathcal{M}$, with $v_1$ connected to all offline vertices in $U$. The advice $A(v_1)$ is chosen arbitrarily. For each subsequent iteration $t = 2, \ldots, n$, the adversary presents online vertex $v_t$, which is adjacent to the neighbors $N(v_{t-1})$ of the previous vertex $v_{t-1}$, excluding two vertices: the previous advice $A(v_{t-1})$ and the offline vertex that has been filled the least so far by $\mathcal{M}$.

Starting from iteration $t = n + 1$, the behaviors of the two adversaries diverge. The robustness adversary $\mathcal{R}$ continues on the vertices advised to be matched so far as in the classical setting of online fractional bipartite matching without advice: each online vertex is adjacent to the same neighbors as the preceding one, except for the offline vertex that has been filled the least so far by $\mathcal{M}$. In contrast, the consistency adversary $\mathcal{C}$ simply presents a matching to the offline vertices that were advised to be unmatched in the first $n$ iterations, allowing the algorithm to fully saturate them.

We formulate a factor-revealing LP that upper bounds the consistency ratio $c$ of any algorithm $\mathcal{M}$ against $\mathcal{C}$ while ensuring the algorithm is $r$-robust against $\mathcal{R}$, for any $r \in [1/2, 1 - 1/e]$. To this end, we assume that $\mathcal{M}$ satisfies the following conditions: $\mathcal{M}$ saturates each online vertex unless its neighbors are all saturated and, in the common phase, $\mathcal{M}$ pushes the same amount to the neighbors except the advised offline vertex at each iteration. We prove these conditions are without loss of generality. The LP is formulated as follows:

$$
\begin{aligned}
\text{maximize } & c \\
\text{subject to } & x_t + (2n - 2t + 1) \cdot \overline{x}_t \leq 1, && \forall t \in \{1, \ldots, n\}, \\
& d_t = \sum_{i=1}^{t-1} \overline{x}_i + x_t, \ \overline{d}_t = \sum_{i=1}^{t} \overline{x}_i, && \forall t \in \{1, \ldots, n\}, \\
& d_t \leq d_{t+1}, && \forall t \in \{1, \ldots, n-1\}, \\
& \sum_{i=t}^{n} y_{i,t} \leq 1, && \forall t \in \{1, \ldots, n\}, \\
& \ell_i^{(t)} = d_i + \sum_{s=1}^{t} y_{i,s}, && \forall t \in \{1, \ldots, n\}, \forall i \in \{t, \ldots, n\}, \\
& \ell_i^{(t)} \leq \ell_{i+1}^{(t)}, && \forall t \in \{1, \ldots, n\}, \forall i \in \{t, \ldots, n-1\}, \\
& \sum_{t=1}^{n} (d_t + \overline{d}_t) + \sum_{t=1}^{n} \sum_{i=t}^{n} y_{i,t} \geq 2nr, \\
& \sum_{t=1}^{n} d_t + n \geq 2nc, \\
& 0 \leq x_t, \overline{x}_t, d_t, \overline{d}_t, y_{i,t}, \ell_i^{(t)} \leq 1, && \forall t \in \{1, \ldots, n\}, \forall i \in \{t, \ldots, n\}.
\end{aligned}
$$

## 6 Experiments

We experimented on synthetic random graphs and real-world graphs. Each plot is generated by letting each algorithm solve 10 instances for 10 different noise parameter values. Source code implementations and experimental scripts are given in the supplementary material.

## 6.1 Graph instances

We experimented on two families of synthetic graphs – Erdős-Rényi (ER) and Upper Triangular (UT), and 6 real-world graphs from the Network Data Repository [RA15]. For $n \in \{100, 200, 300\}$ and edge probability $p \in \{0.1, 0.2, 0.5\}$, ER$(n, p)$ graphs are generated with $n$ offline and $n$ online vertices, with each possible bipartite edge existing independently with probability $p$. For $n \in \{100, 200, 300\}$, each UT$(n)$ graph consists of $n$ offline vertices and $n$ online vertices, where the $i$-th online vertex is connected to the last $n - i + 1$ offline vertices. Meanwhile, we pre-process[6] real-world graphs in a similar manner to [BKP20] to obtain random bipartite graphs: first, shuffle all $n$ vertices indices in the real-world graph, take the first $\lfloor n/2 \rfloor$ as the offline vertices and the next $\lfloor n/2 \rfloor$ as online vertices and only keep the bipartite crossing edges. For weighted instances, each offline vertex is given a random weight between 0 and 1000.

## 6.2 Advice generation

For each graph $\mathcal{G}$ with $n$ vertices and a given noise parameter $\gamma \in [0, 1]$, we generate a noisy prediction $\widehat{\mathcal{G}}_\gamma$ of $\mathcal{G}$ as follows: each online vertex $v$ retains a random $(1 - \gamma)$ fraction of its true neighbors and gains a random $\gamma$ fraction of its non-neighbors. Thus, when $\gamma = 0$, the prediction is exact ($\widehat{\mathcal{G}}_0 = \mathcal{G}$), and when $\gamma = 1$, it corresponds to the complement graph ($\widehat{\mathcal{G}}_1 = \overline{\mathcal{G}}$).

To generate the advice for the $t$-th arriving online vertex (for $t \in [n]$), we solve a linear program that maximizes the (weighted) matching objective. This is done subject to two components: the actual decisions made for the first $t - 1$ arrivals in the true graph $\mathcal{G}$, and a noisy prediction of the future arrivals from time $t + 1$ to $n$, based on $\widehat{\mathcal{G}}_\gamma$. Importantly, the current arrival at time $t$ is not included in the noisy future but is instead the decision variable of interest. In more detail, the advice at time $t$ is generated by perturbing the true future subgraph (i.e., the part of $\mathcal{G}$ involving vertices $t + 1$ to $n$) to create a noisy forecast. We then solve for the optimal decision at time $t$ that maximizes the matching value, given the past decisions up to $t - 1$ (in $\mathcal{G}$) and the predicted future (in $\widehat{\mathcal{G}}_\gamma$). Since we use the true graph up to and including time $t$, this process ensures that the advice at each time step is always feasible and based on a valid optimization problem over a fully specified $n$-vertex instance.

## 6.3 Benchmarked algorithms

The two baselines are GREEDY and BALANCE. The former greedily matches the online vertex with its highest weighted available offline neighbor while the latter fractionally matches based on the penalty function $g(z) = \exp(z - 1)$. In the unweighted setting, BALANCE is equivalent to the classic WATERFILLING algorithm. Note that both GREEDY and BALANCE are independent of any predictions, so they achieve constant performance for any noise parameter $\gamma \in [0, 1]$. We also implemented and benchmarked our LAB and PAW algorithms, which take as inputs $\lambda_{\text{LAB}}$ and $\lambda_{\text{PAW}}$ respectively. Note that we only run PAW for unweighted instances. Recall from Theorems 1 and 3 that LAB and PAW have different consistency values with respect to their parameters: the consistency of LAB is $1 + \lambda_{\text{LAB}} - \exp(\lambda_{\text{LAB}} - 1)$ while the consistency of PAW is $1 - (1 - \lambda_{\text{PAW}}) \exp(\lambda_{\text{PAW}} - 1)$. To compare between them at the same consistency value, we set $\lambda_{\text{PAW}} = 1 + W(\lambda_{\text{LAB}} - \exp(\lambda_{\text{LAB}} - 1))$. Since BALANCE already achieves a competitive ratio of $1 - 1/e \approx 0.632$, we consider consistency ratios of $\{0.7, 0.8, 0.9, 1.0\}$ when running LAB and PAW. For consistency ratio of 0.7, we set $\lambda_{\text{LAB}} \approx 0.11$ and $\lambda_{\text{PAW}} \approx 0.51$. For consistency ratio of 0.8, we set $\lambda_{\text{LAB}} \approx 0.29$ and $\lambda_{\text{PAW}} \approx 0.74$. For consistency ratio of 0.9, we set $\lambda_{\text{LAB}} \approx 0.52$ and $\lambda_{\text{PAW}} \approx 0.89$. For consistency ratio of 1.0, we set $\lambda_{\text{LAB}} = 1$ and $\lambda_{\text{PAW}} = 1$.

## 6.4 Qualitative takeaways

Fig. 7 illustrates a subset of our empirical results. As predicted by our analysis, the competitive ratio attained by both LAB and PAW degrades as the noise parameter $\gamma$ increases. In particular, when $\gamma = 0$ (i.e., perfect advice), both LAB and PAW achieve a competitive ratio of 1 when $\lambda_{\text{LAB}} = \lambda_{\text{PAW}} = 1$. As $\gamma$ grows large, the advice becomes increasingly uninformative, and it is unsurprising that the advice-free algorithm BALANCE eventually outperforms both learning-augmented algorithms, with the crossing point depending on the underlying graph instance.

---

[6]Such a pre-processing step is necessary because these real-world graphs are not bipartite to begin with.

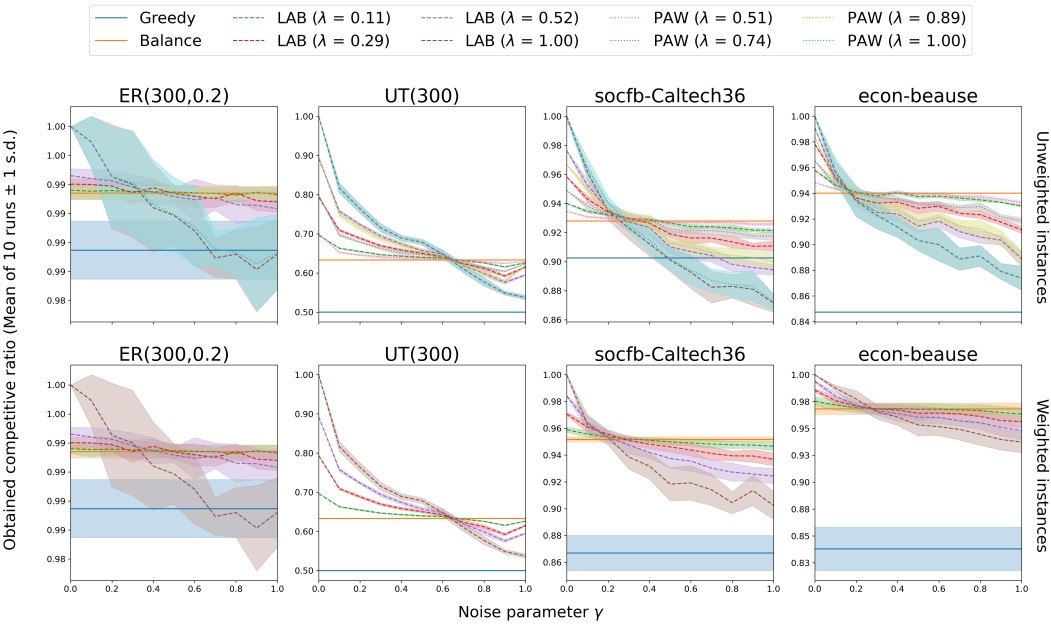

Figure 7: Subset of empirical results: ER(300, 0.2), UT(300), and 2 real-world graphs (socfb-Caltech36, econ-beause). See the supplementary material for our full set of experiments.

Interestingly, across all our experiments — including those in the appendix — we consistently observe a phenomenon where there appears to exist a critical noise level $\gamma^*$ such that the competitive ratios of all runs of LAB and PAW (across different $\lambda$ values) converge and coincide with that of BALANCE. This suggests that at $\gamma^*$, the advice becomes effectively uncorrelated with the input, causing the behavior of LAB and PAW to resemble that of BALANCE regardless of the weighting parameter $\lambda$. While we do not currently have a theoretical explanation for this convergence, it is a compelling empirical observation that may point to deeper structure in the robustness-consistency tradeoff and warrants further investigation in future work.

## 7 Conclusion and Open Problems

We studied the robustness-consistency tradeoffs of learning-augmented algorithms for online bipartite fractional matching. We proposed and analyzed two algorithms, LAB and PAW, and established an improved hardness result.

In our current work, PAW relies on integral advice while LAB can accommodate fractional advice. While it is a natural question to ask if there can be a unified algorithm and analysis, our current analytical framework is unable to do so. The analysis of LAB is agnostic to the weights, making it unclear how to demonstrate an improved tradeoff in the unweighted case. Meanwhile, the analysis of PAW crucially relies on the integrality of the advice, and we were unable to obtain a comparable bound in the fractional case. We do not rule out the possibility of a unified analysis, and we view this as a compelling direction for future work. We do not rule out the possibility of a unified analysis and view this as an intriguing direction for future work.

Besides unifying the two variants, there are serveral other natural open and interesting research directions. Firstly, it would be interesting to develop a theoretical explanation for the crossing point phenomenon observed in our experiments; see the discussion in Section 6.4. Another direction would be to close the gap between our algorithmic results and the impossibility bounds. Progress on this front could come from an analytic proof of the impossibility result, as well as a tight analysis of LAB in the unweighted setting. Finally, it would be interesting to extend our framework to broader variants of online matching, including Display Ads, the generalized assignment problem [FKM+09, SE23], and the multi-stage setting [FN24].

## Acknowledgments and Disclosure of Funding

This research/project is supported by the National Research Foundation, Singapore under its AI Singapore Programme (AISG Award No: AISG-PhD/2021-08-013). This work was partly supported by Institute of Information & communications Technology Planning & Evaluation (IITP) grant funded by the Korea government (MSIT) (No. RS-2021-II212068, Artificial Intelligence Innovation Hub). This work was partly supported by an IITP grant funded by the Korean Government (MSIT) (No. RS-2020-II201361, Artificial Intelligence Graduate School Program (Yonsei University)). Supported by NCN grant number 2020/39/B/ST6/01641.

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

# A Pseudocodes of our algorithms

---

**Algorithm 1:** Learning-Augmented Balance Algorithm (LAB)

---

**Input:** Offline vertices $U$, tradeoff parameter $\lambda \in [0, 1]$
**Data:** Online vertices $V$, edges $E$, and fractional advice $a \in \mathbb{R}^E$
**Output:** Fractional matching $x \in \mathbb{R}^E$

1   **foreach** $u \in U$ **do**
2      $X_u \leftarrow 0$                                      // Amount allocated by algorithm
3      $A_u \leftarrow 0$                                        // Amount allocated by advice
4   **foreach** *arrival of $v \in V$ with neighbors $N(v)$ and advice $\{a_{u,v}\}_{u \in N(v)}$* **do**
5      **foreach** $u \in N(v)$ **do**
6          $A_u \leftarrow A_u + a_{u,v}$                               // Accumulate advice
7
8      Find the smallest $\ell \geq 0$ such that $\sum_{u \in N(v)} x_{u,v} \leq 1$, where
         $x_{u,v} := \min\{z \in [0, 1 - X_u] \mid w_u \cdot (1 - f(A_u, X_u + z)) \leq \ell\}$    // e.g. via binary search
9
10     **foreach** $u \in N(v)$ **do**
11       $X_u \leftarrow X_u + x_{u,v}$                         // Accumulate actual fractional matching
12
13   **return** $x$

---

---

**Algorithm 2:** Push-and-Waterfill Algorithm (PAW)

---

**Input:** Offline vertices $U$, trade-off parameter $\lambda \in [0, 1]$
**Data:** Online vertices $V$, edges $E$, and integral advice $A : V \to U \cup \{\bot\}$
**Output:** Fractional matching $x \in \mathbb{R}^E$

1   **foreach** $u \in U$ **do**
2      $d_u \leftarrow 0$                                        // Level of $u$
3   **foreach** *arrival of $v \in V$ with neighbors $N(v)$ and advice $A(v)$* **do**
4      **(Phase 1)**: Push to advised neighbor $A(v)$, up to $\tau = \max\{0, \lambda - d_{A(v)}\}$ amount
5      **if** $A(v) \in N(v)$ **then**
6         $\tau \leftarrow \max\{0, \lambda - d_{A(v)}\}$
7         $x_{A(v),v} \leftarrow \tau$
8         $d_{A(v)} \leftarrow d_{A(v)} + \tau$
9      **else**
10        $\tau \leftarrow 0$
11     **(Phase 2)**: Waterfill the remaining $1 - \tau$
12     Find the largest $\ell$ such that $\sum_{u \in N(v)} \max\{0, \ell - d_u\} \leq 1 - \tau$
13     $\ell \leftarrow \min\{\ell, 1\}$
14     **foreach** $u \in N(v)$ **do**
15       $x_{u,v} \leftarrow x_{u,v} + \max\{0, \ell - d_u\}$
16       $d_u \leftarrow d_u + \max\{0, \ell - d_u\}$
17   **return** $x$

---

