# OpenReview forum: "Learning-Augmented Online Bipartite Fractional Matching"
_NeurIPS.cc/2025/Conference — NeurIPS 2025 poster_

### Official Review · Reviewer_EhSo · 2025-06-20

**Clarity:** 3
**Significance:** 3
**Originality:** 3
**Rating:** 4
**Confidence:** 4

**Summary:**

This paper studies online bipartite fractional matching with advice and proposes two learning-augmented algorithms—LAB for vertex-weighted and PAW for unweighted settings—that provably outperform existing methods. Both algorithms dominate the naive COINFLIP baseline across the entire robustness-consistency tradeoff, a first in this line of work. LAB further extends to the AdWords problem under the small bids assumption, improving upon classic results by Mahdian et al. The paper also establishes new theoretical upper bounds by constructing hard instances that reveal fundamental limits of any algorithm in this framework. Extensive experiments on synthetic and real-world graphs confirm the algorithms’ effectiveness: they achieve optimal performance under accurate advice and gracefully degrade as advice quality worsens, aligning with theoretical predictions. This work advances the theory and practice of learning-augmented online optimization.

**Questions:**

No.

**Ethical Concerns:**

["NO or VERY MINOR ethics concerns only"]

**Final Justification:**

This paper presents a thorough and impactful contribution to the field of online optimization.

**Limitations:**

Yes

**Paper Formatting Concerns:**

No.

**Quality:**

3

**Strengths And Weaknesses:**

Strengths and Weaknesses:

This paper presents a thorough and impactful contribution to the field of online optimization. It introduces two learning-augmented algorithms, LAB and PAW, for online bipartite fractional matching in both vertex-weighted and unweighted settings. A key strength is the theoretical breakthrough: both algorithms provably dominate the COINFLIP baseline across the entire robustness-consistency tradeoff spectrum—a significant advancement over prior works, which only achieved partial domination. Additionally, the extension of LAB to the AdWords problem under the small bids assumption enhances its practical applicability, improving upon the classic Mahdian-Nazerzadeh-Saberi framework. The authors further provide tight upper bounds on what any algorithm can achieve via a novel hardness construction, and back up their theoretical findings with comprehensive experiments on both synthetic and real-world datasets, demonstrating smooth performance degradation with increasing noise in the advice.

However, the paper’s technical exposition could be improved. While the core algorithmic ideas and analysis are sound, several important definitions and procedural steps are presented in a terse or abstract way, making the work less accessible to readers unfamiliar with the topic. For example, the intuition behind the penalty functions and dual constructions could be elaborated with more concrete examples or illustrations. Strengthening the clarity and pedagogical structure of the technical sections—particularly the algorithm descriptions and analytical proofs—would make the paper more readable and broaden its impact beyond the expert audience.

---

> ### Author Rebuttal · Authors · 2025-07-30
>
> Thank you for your time and effort in reviewing our work, and for providing useful feedback, especially about a better exposition of our paper.
>
> We will revise our paper to give more intuition as follows:
> 1. Regarding the penalty functions, we mentioned in the full version an intuition on $f(A,X)$ for fractional advice; unfortunately, $f_0(X)$ and $f_1(X)$ for integral advice are mechanically derived from our primal-dual analysis, so we have little intuition for the definition of these functions. Including this discussion in the main version would make the readers better understand our penalty functions.
> 2. Regarding our algorithm descriptions and dual constructions, we believe that pseudocodes incorporating the dual construction would help the readers understand our algorithms and dual constructions more clearly. We already provided pseudocodes of our algorithms without dual construction in the appendix of our full version, but we can move them to the main body and add the dual construction part in the revision.
> 3. We agree that our analysis for LAB is quite long, and hence, it would be difficult for the readers to follow the analysis in their first reading. We will add an overview of the analysis in the revision.

---

### Official Review · Reviewer_cUNG · 2025-06-22

**Clarity:** 3
**Significance:** 2
**Originality:** 2
**Rating:** 3
**Confidence:** 4

**Summary:**

The paper under consideration studies the vertex-weighted online bipartite fractional matching problem with predictions. That is, there is a bipartite graph where one side is available to the algorithm upfront offline. Every offline vertex is assigned a weight (if all weights coincide, the instance reduces to an unweighted one). The other side of vertices arrives online one-by-one.
In the purely adversarial setting, nodes arrive without any information and the decision maker needs to immediately and irrevocably decide to which extend an edge is (fractionally) added to the matching.
In the setting with predictions, there is a prediction for each online vertex indicating to which extend each incident edge should be considered.

The paper derives algorithms for the unweighted and vertex-weighted setting and derives results for robustness and consistency---a classical approach in the algorithms with predictions literature. More precisely, for the vertex-weighted variant, a learning-augmented version of the balance algorithm is used to derive the robustness-consistency guarantees. For the unweighted case, an extension of the classical waterfilling algorithm is developed.

From a technical perspective, the analysis is based on a primal-dual approach. In particular, depending on the arrivals, primal and dual variables are updated in a suitable way to maintain feasible solutions for both problems. These can then be used to apply typical primal-dual arguments for competitive ratios.

The authors can extend their results also to the AdWors problem when imposing a small-bid assumption and complement results with an upper bound for the integral-advice case.

**Questions:**

* One of my main concerns is the novelty and originality of the paper: Even though I like the general approach, I got the impression that this work only adds marginal value to the huge body of literature in the algorithms-with-predictions regime. Can you clearly state the main differences why this work is adding sufficiently compared to previous work, please? (In the paper, there is only a small paragraph on page 2 on this unfortunately)

* In the paper, you are required to use two algorithms for the vertex-weighted and unweighted case. Do you see a chance of unifying both settings? More precisely, what is the deeper intuition why these two approaches are required?

* As a follow-up point, can it be the case that the primal-dual analysis is simply too lossy to get reasonable guarantees in both cases?

**Ethical Concerns:**

["NO or VERY MINOR ethics concerns only"]

**Final Justification:**

I keep my score.

**Limitations:**

No negative societal impact.

**Paper Formatting Concerns:**

The paper is well formatted and written, hence nice and easy to read.

**Quality:**

3

**Strengths And Weaknesses:**

*Strengths*:
The paper derives reasonable guarantees for robustness and consistency in a variety of settings in online bipartite fractional matching. In particular, the guarantees outperform trivial bounds from coinflip arguments and are extendable to more general settings (as AdWords, with a few reasonable assumptions)---indicating the power of this approach. The primal-dual analysis is nice and clean, making it an enjoyable read.

*Weaknesses*:
That said, the paper is required to use different algorithms for the different problem formulations---tailoring the approach to the respective problem. The design of the algorithms feels natural, but not really surprising. Given this design, the primal-dual approach for the analysis is expectable.
Having a unified approach for both cases (unweighted and vertex-weighted) would be a very desirable goal.

---

> ### Author Rebuttal · Authors · 2025-07-30
>
> Thank you for your time and effort in reviewing our work, and for providing useful feedback.
>
> You are correct that PAW requires integral advice while LAB can handle fractional advice, and we agree with you that it would be nice if there is a way to unify the two. Unfortunately, this is a limitation of our current analytical approach: our analysis for LAB is independent from the weights, and so it is not obvious to us how to show that LAB attains an improved tradeoff in the unweighted setting. On the other hand, the analysis of PAW depends on the integrality of the advice, and we were unable to find the same bound for the fractional case. We do not rule out the possibility that there is a way to unify the two, and we view this as an interesting open problem to resolve. We will add a clearer discussion about PAW versus LAB in our revision, and motivate this open problem more clearly.
>
> In terms of contribution, we would like to respectfully disagree with the reviewer's view that our work is marginal. Online bipartite matching is a fundamental problem in Computer Science and Operations Research that has been studied extensively. Given the increasing availability of machine learning and AI prediction tools, it is important that the research community studies how such predictions can interface with classic problems to yield provable improvements. Learning-augmented algorithms provide such a principled framework for doing so. However, all prior work that studies the problem of learning-augmented online bipartite matching either considered a restricted special case or did not even beat the performance curve obtained by the naive coin-flip algorithm (see Figure 1 and the surrounding discussion). As such, there was a big gap in our understanding of the problem, and our paper contributes towards filling this gap.
>
> As the primal-dual technique has proven successful in providing tight analyses for many variants of online matching, we do not currently have reasons to believe that the framework is inherently "too lossy". We believe that the failure to unify the two analyses is an artefact of our current approach.

---

> > ### Comment · Reviewer_cUNG · 2025-08-04
> >
> > I thank the authors for their response. Given the other positive reviews, I will not vote against accepting this paper at NeurIPS.

---

### Official Review · Reviewer_zQgB · 2025-06-26

**Clarity:** 3
**Significance:** 3
**Originality:** 3
**Rating:** 5
**Confidence:** 3

**Summary:**

This paper tackles the problem of online bipartite fractional matching with the support of advice. In this setting, there exists a bipartite graph between a set of offline vertices $U$ that have a value associated with them and a set of online vertices $V$ that arrive one by one. Upon the arrival of each online vertex, its neighbors along with suggested fractional matchings to its neighbors are revealed. The algorithm must then decide, in real time and without future knowledge, how to match (fractionally) the arrived online vertex to its neighbors subject to feasibility constraints.

The authors present two novel algorithms:

* LEARNINGAUGMENTEDBALANCE (LAB) for the vertex-weighted setting, which adapts the classic BALANCE algorithm using an advice-aware penalty function.
* PUSHANDWATERFILL (PAW) for the unweighted setting, which introduces a two-phase approach combining targeted flow pushing and traditional waterfilling.

A key contribution is that both algorithms strictly improve upon the baseline COINFLIP algorithm across the entire robustness-consistency tradeoff spectrum, which quantifies the balance between worst-case guarantees and performance under accurate advice.

The LAB algorithm is also shown to improve upon prior work for the AdWords problem under the small bids assumption, offering better guarantees than the well-known Mahdian et al. framework. On the other hand, PAW is tailored to overcome analysis challenges specific to the unweighted setting and shows strong performance both theoretically and empirically.

Additionally, the paper provides a new upper bound on the robustness-consistency tradeoff, which is shown to be tight in many cases and improves over previous results. Experimental evaluations on both synthetic and real-world bipartite graphs demonstrate that the proposed algorithms interpolate smoothly between advice-based and advice-free strategies, with performance aligning with theoretical predictions.

**Questions:**

* In each experiment, it seems like there exists a special value $\gamma^*$ such that at that level of noise, the CR of all executions of LAB and PAW (with different values of $\lambda$) and the BALANCE algorithm coincide. This can be seen in the experiments in the appendix as well. Could the authors clarify if there is a theory behind this observation and whether this convergence has theoretical significance? Is this due to advice being entirely uncorrelated at that γ, leading LAB to behave equivalently to BALANCE regardless of λ?

* Can the authors clarify the reason why the PAW algorithm does not extend to the case where the advice is also fractional?

**Ethical Concerns:**

["NO or VERY MINOR ethics concerns only"]

**Final Justification:**

My concerns regarding the experimental setup are resolved.

**Limitations:**

Yes.

**Paper Formatting Concerns:**

Instead of "an appendix of the paper", the supplemental material contains a full version of the paper with much more details. Because of that, it is not easy to read the main text and only refer to the "appendix" for the missing proofs or details. I either had to read the main text (with no proof) or read the full version. For instance, Lemma 6 in the main text appears as Lemma 9 in the full paper and Lemma 7 appears as Lemma 12 in the full paper provided as an "appendix", with some notations and observations provided in the text preceding the lemma in the full version. Anyway, I don't think this is a "violation" of paper formatting criteria.

**Quality:**

3

**Strengths And Weaknesses:**

**Strengths**
* The paper presents two new learning-augmented algorithms: LAB, which modifies the BALANCE algorithm with advice-aware penalties, and PAW, which adds a controlled "push" step to WATERFILLING for unweighted graphs, achieving better tradeoffs than COINFLIP.
* The paper develops a factor-revealing LP to derive hardness results, improving prior bounds.
* Experiments across a variety of datasets and noise levels confirm theoretical insights.

**Weaknesses**
* There is an assumption that the advice to the PAW algorithm is integral, while the algorithm generates fractional matchings.
* I have some concerns with the experimental setup. In particular, I like the idea of having a parameter $\gamma$ controlling the accuracy of the provided advice; however, the same level of noise is not reflected in the CR of the algorithms simply because the advice matches the online vertices to non-neighbor offline vertices, which are infeasible and the LAB algorithm corrects the infeasibility by matching to other points. (To better understand my concern, see Figure 4: when $\lambda=1$ (the LAB fully trusts the advice and completely follows it) and $\gamma=1$ (fully noisy setting where advice is completely irrelevant), the CR is still above 0.9 in three cases). I believe the experiments are more meaningful if the algorithm is tested on valid but poor advice. For instance, I expect to see a CR of 0.63 or so when $\lambda=1$ and the advice provides a 1/2-approx matching.

---

> ### Author Rebuttal · Authors · 2025-07-30
>
> Thank you for your time and effort in reviewing our work, and for providing useful feedback.
>
> You are correct that PAW requires integral advice while LAB can handle fractional advice, and we agree with you that it would be nice if there is a way to extend the analysis of LAB to handle the unweighted case. Unfortunately, this is a limitation of our current analytical approach: our analysis for LAB is independent from the weights, and so it is not obvious to us how to show that LAB attains an improved tradeoff in the unweighted setting. On the other hand, the analysis of PAW depends on the integrality of the advice, and we were unable to find the same bound for the fractional case. We do not rule out the possibility that there is a way to unify the two, and we view this as an interesting open problem to resolve. We will add a clearer discussion about PAW versus LAB in our revision, and motivate this open problem more clearly.
>
> As for the experimental concern, we would like to point out that the robustness ratio is a provable guarantee that holds for any instance (even worst-case ones) while your concerns of high competitive ratio were referring to random ER (Erdos-Renyi) instances and real-world instances (which are unlikely to be worst-case for our proposed algorithms). The trend you are expecting happens in the synthetic UT (upper-triangular) instances. We would also like to highlight that our advice is always valid: when generating advice, perturbation is only applied for the prediction of the future (excluding current time step), so the advice will only be non-zero on actual neighbors of the arriving vertex.
>
> Regarding your question about coinciding intersection points: Unfortunately, we do not have a working theory explaining this phenomenon but it is an interesting observation that may warrant further investigation in future work. We will mention this phenomenon in our revision.
>
> Finally, for the paper formatting concern, we didn’t notice such inconvenience when submitting our paper. We will fix this issue in the camera-ready version by, for example, matching the numbering of the main paper with the full version. We apologize for the inconvenience and thank you very much for pointing out this issue.

---

> > ### Comment · Reviewer_zQgB · 2025-08-01
> >
> > I thank the authors for their complete response.
> >
> > Concern with the experiments: As in line 282 of your paper, the advice for each time t is generated by solving an LP for the first t-1 arrivals in the original graph and the remaining n-t arrivals on the noisy graph. So the advice for the t-th arrival is given based on the edges on the noisy graph, unless there is a typo in the writing or I misunderstood that part.

---

> > > ### Author Response · Authors · 2025-08-01
> > >
> > > Thank you for the prompt response and follow-up. To clarify, the way we generate the advice at time $t$ is as follows: we take the true future graph and apply a perturbation to it. Then, using this perturbed graph (which we think of as a forecast of the true future), we solve for the decision at time t that maximizes the value of the matching, given the decisions the algorithm has made up to that point (from the first $t-1$ arrivals). Note that the total number of arrivals is $(t-1) + 1 + (n-t) = n$, and the "1" is the current arrival that is *excluded* from the noisy final $n-t$ arrivals.  In this way, the advice at time t is always a valid decision, since we have access to the true graph up to and including time $t$ when generating it. We will add a version of this clarification in our revision. Please let us know if you have further questions!

---

> > > > ### Comment · Reviewer_zQgB · 2025-08-03
> > > >
> > > > I thank the authors for their response. I raised my score to accept.

---

### Official Review · Reviewer_g5iY · 2025-06-30

**Clarity:** 4
**Significance:** 3
**Originality:** 3
**Rating:** 5
**Confidence:** 4

**Summary:**

In this paper, the authors consider online bipartite fractional matching problems when the online decision maker is provided advice in the form of a suggested (offline feasible) matching in each iteration. They design algorithms for both the vertex-weighted and unweighted versions of the problem that are provably better than the coin-flipping strategy of randomly choosing between advice-following and advice-free algorithms. They also show that the vertex-weighted version can be extended to the AdWords problem with small bids. They also provide hardness bound on the robustness-consistency tradeoff and validate their algorithms through synthetic and real-world data.

**Questions:**

The authors list several interesting open questions in their conclusion. I wonder if they have any comments indicating how some of the content and technical ideas within the paper could be extended or be helpful (or not) in addressing these questions.

**Ethical Concerns:**

["NO or VERY MINOR ethics concerns only"]

**Final Justification:**

I am fully supportive for this paper to being accepted

**Limitations:**

yes

**Quality:**

3

**Strengths And Weaknesses:**

Strengths:
- Results, claims and proofs are technical sound and solid and provide a good picture on what can be achieved for these problems under learning-augmented algorithms.
- Paper is very well written, without going into unnecessary technical details within the main body.
- Content of the paper provides additional and novel aspects to the literature on online bipartite fractional matching problems under learning-augmented algorithms.
- Novel and well positioned paper.

Weaknesses:
- Would have been nice to get additional insights on the open problems listed in the conclusion, but this is a minor weakness, as I liked the paper ...

---

> ### Author Rebuttal · Authors · 2025-07-30
>
> Thank you for your time and effort in reviewing our work, and for providing useful feedback.
>
> Many of the existing works that study the problems mentioned in our conclusion are based on primal-dual analysis. As such, we hope that our learning-augmented primal-dual technique proposed in this paper extends to these settings too.

---

> > ### Comment · Reviewer_g5iY · 2025-08-04
> >
> > I have looked at all reviews and rebuttals by the authors. I will maintain my positive score.

---

### Official Review · Reviewer_ANVd · 2025-07-03

**Clarity:** 4
**Significance:** 4
**Originality:** 4
**Rating:** 5
**Confidence:** 5

**Summary:**

This paper studies learning-augmented algorithms for online bipartite *fractional* matching, aiming to outperform the COINFLIP baseline by achieving better robustness–consistency tradeoffs.

In online bipartite matching, a bipartite graph is revealed incrementally: the offline vertices are known in advance, while online vertices arrive one by one, requiring irrevocable matching decisions to maximize the total matched weight. (In the vertex-weighted case, this means maximizing the sum of the offline vertex weights.)

The work focuses on a predictive setting where, upon each arrival, the algorithm receives advice suggesting a fractional matching. The algorithm seeks to remain robust to poor advice while exploiting accurate predictions.

The authors propose two algorithms: LAB for the weighted case and PAW for the unweighted case, and establish hardness results that delineate the limits of such tradeoffs.


The COINFLIP algorithm randomly chooses between an advice-free strategy that guarantees worst-case performance and an advice-following strategy that assumes the predictions are perfectly accurate, effectively tracing a linear tradeoff curve between robustness and consistency.
They show that there exists a learning-augmented algorithm for online bipartite matching that dominates COINFLIP across the entire range of robustness, providing improved guarantees regardless of how accurate the advice is.

They mainly use a primal-dual analysis framework to design and analyze their algorithms, adapting classical techniques to incorporate advice-dependent penalty functions that balance following predictions and maintaining worst-case guarantees.


For the vertex-weighted case, they use a version of a continuous process that incrementally pushes flow to neighbors based on an advice-aware carefully chosen penalty function, prioritizing offline vertices with higher weights and those recommended by the advice.

For the unweighted case, they propose a different algorithm called PUSHANDWATERFILL, which first pushes flow to the advised offline vertex up to a specified threshold and then distributes any remaining flow using the standard waterfilling strategy, allowing better robustness-consistency tradeoffs than COINFLIP in this setting.

They also improve the upper bound by designing a hardness construction using two adaptive adversaries, one targeting robustness and the other consistency, which remain indistinguishable during the initial phase, making it provably impossible for any algorithm to exceed certain tradeoffs across the entire spectrum.




The COINFLIP algorithm randomly chooses between an advice-free strategy that guarantees worst-case performance and an advice-following strategy that assumes the predictions are perfectly accurate, effectively tracing a linear tradeoff curve between robustness and consistency.
They show that there exists a learning-augmented algorithm for online bipartite matching that dominates COINFLIP across the entire range of robustness, providing improved guarantees regardless of how accurate the advice is.

They mainly use a primal-dual analysis framework to design and analyze their algorithms, adapting classical techniques to incorporate advice-dependent penalty functions that balance following predictions and maintaining worst-case guarantees.


For the vertex-weighted case, they use a version of a continuous process that incrementally pushes flow to neighbors based on an advice-aware carefully chosen penalty function, prioritizing offline vertices with higher weights and those recommended by the advice.

For the unweighted case, they propose a different algorithm called PUSHANDWATERFILL, which first pushes flow to the advised offline vertex up to a specified threshold and then distributes any remaining flow using the standard waterfilling strategy, allowing better robustness-consistency tradeoffs than COINFLIP in this setting.

They also improve the upper bound by designing a hardness construction using two adaptive adversaries, one targeting robustness and the other consistency, which remain indistinguishable during the initial phase, making it provably impossible for any algorithm to exceed certain tradeoffs across the entire spectrum.

**Questions:**

None.

**Ethical Concerns:**

["NO or VERY MINOR ethics concerns only"]

**Final Justification:**

I have decided to maintain a positive score for this paper.

**Limitations:**

Yes.

**Paper Formatting Concerns:**

None.

**Quality:**

4

**Strengths And Weaknesses:**

This paper makes a strong contribution to learning-augmented online optimization by introducing algorithms that achieve improved robustness-consistency tradeoffs across the entire spectrum. The authors address one of the most fundamental problems in online algorithms—online bipartite matching—combining elegant algorithm design with rigorous primal-dual analysis and an improved hardness construction that clarifies the theoretical limits of the problem. Empirical results on diverse datasets convincingly support the theoretical claims.

I recommend accepting this paper as it advances both the theoretical foundations and practical understanding of learning-augmented online matching.

---

> ### Author Rebuttal · Authors · 2025-07-30
>
> Thank you for your time and effort in reviewing our work, and for appreciating our contributions.

---

### Decision · Program_Chairs · 2025-09-17

**Decision:**

Accept (poster)

**Comment:**

Learning augmented algorithms (algorithms with hints) are a bit niche for NeurIPS, but this is a very interesting paper that received reviews from the LLA community and the ML ones.

The former are more excited than the second ones, but that was expected.

I personnally believe that LLA is a very promising research direction for the ML community (not just TCS), thus I am happy to support acceptance of this paper, as most of the reviewers.